# FGD-VO: Flow-Guided Deformable Correlation with Hybrid Masks for Visual Odometry

## Abstract

Learning-based visual odometry remains vulnerable to outliers, particularly dynamic objects and background regions irrelevant to motion. It also suffers from weak modeling of inter-frame geometry, that is, the explicit pixel-wise correspondences that provide reliable motion cues. Many methods still rely on frame concatenation followed by convolutions, which tends to overlook fine-grained pixel correspondences across frames. This paper presents FGD-VO, an end-to-end monocular VO framework that addresses these issues within a unified architecture. First, we introduce a Flow-Guided Deformable Correlation (FGDC) module, which leverages dense optical flow to locate pixel correspondences between consecutive frames and augments them with learnable local offsets, enabling the aggregation of geometrically relevant features beyond fixed pixel matches. Second, we propose a Hybrid Masking strategy that combines an explicit optical-flow consistency mask with an implicit, learnable attention mask, allowing the network to simultaneously suppress unreliable correspondences and adaptively emphasize informative regions during pose refinement. Extensive experiments on the KITTI odometry benchmark demonstrate that FGD-VO achieves state-of-the-art accuracy among learning-based VO methods, significantly reducing both translational and rotational errors. Our findings suggest that explicitly coupling flow-guided deformable correlation with hybrid masking is a promising direction for improving the reliability and generalization of real-time visual odometry in autonomous systems. We will release our source code to facilitate reproducibility and future research.

## 1 INTRODUCTION

Visual odometry (VO) is a core problem in computer vision and robotics, which estimates the motion trajectory of an agent by analyzing consecutive images from onboard cameras, providing essential information for localization and navigation. Traditional VO methods typically estimate motion by matching hand-crafted features—such as SIFT(Lowe, 2004) or ORB(Rublee et al., 2011)—to infer camera movement, or by directly minimizing photometric discrepancies across pixel intensities to recover relative pose(Newcombe et al., 2011; Engel et al., 2014; 2017). Although traditional VO approaches generally deliver high accuracy, they struggle in texture-sparse regions such as tunnels, corridors, or nighttime scenes, and in highly dynamic environments like urban traffic with dense vehicles and pedestrians, which undermine their robustness in real-world applications. In recent years, deep learning has been applied to visual odometry by treating pose estimation as a regression task. Early work relied on convolutional neural networks (CNNs) to predict relative camera motion directly from consecutive image pairs(Costante & Ciarfuglia, 2018). To capture temporal consistency, later approaches introduced recurrent units—such as LSTM-based RNNs—to aggregate information over multiple frames and smooth trajectory predictions(Wang et al., 2018). More recently, Transformer-inspired architectures have been explored, leveraging self-attention to model long-range spatio-temporal relationships and further enhance pose accuracy(Françani & Maximo, 2025).

Despite recent progress, deep learning–based visual odometry still faces two major challenges. First, outliers caused by occlusions, large viewpoint changes, or dynamic objects (e.g., overtaking vehicles in highway scenarios) often introduce spurious motion cues, leading to trajectory drift or jitter. Second, insufficient modeling of inter-frame geometry makes existing approaches vulnerable in cases of wide-baseline motion or low-texture areas, where reliance on appearance features alone fails to ensure stable alignment.

For the first challenge, existing literature commonly addresses the impact of dynamic objects and mismatches by constructing explicit consistency masks using optical flow and/or depth estimation. In this paradigm, dense optical flow provides pixel correspondences between consecutive frames, and regions with large motion inconsistencies are assigned low confidence or excluded from pose estimation(Yin & Shi, 2018; Bian et al., 2019; Kuo et al., 2020; Cho & Kim, 2023; Li et al., 2025). While effective at rejecting gross mismatches, flow-consistency masks still allow through many residual outliers. For instance, fast-moving vehicles that momentarily align across frames can severely bias motion estimation. This rigid thresholding also lacks adaptability, often discarding useful pixels in challenging conditions such as partial occlusions. To address this limitation, our method integrates the explicit optical-flow consistency mask with a learnable, implicitly predicted mask that is iteratively refined during the pose estimation process, enabling the network to adaptively suppress residual non-informative regions while emphasizing features most relevant to camera motion.

For the second challenge, many prior works address inter-frame geometry by concatenating two input images and applying convolutional networks for joint feature extraction. This design implicitly assumes that pixels at the same coordinates in consecutive frames correspond to the same 3D point, an assumption that rarely holds in practice due to camera motion and scene geometry (Figure 1a) (Wang et al., 2017; Ummenhofer et al., 2017; Yin & Shi, 2018; Hwang et al., 2022; Zhao et al., 2022; Bian et al., 2019). As a result, fea-

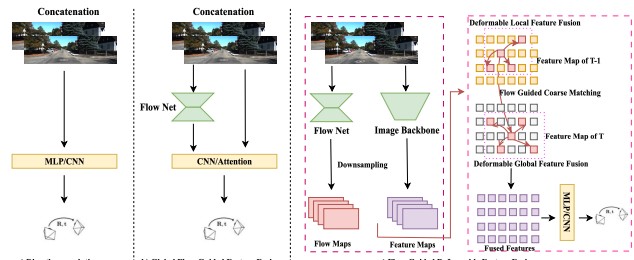

Figure 1: Comparison of different feature fusion strategies for visual odometry. (a) Direct concatenation of consecutive images followed by convolution. (b) Global fusion with optical flow guidance. (c) Proposed flow-guided deformable feature fusion, which combines coarse correspondence estimation from optical flow with learnable local offsets for fine-grained alignment.

tures from unrelated spatial regions can be mixed, weakening the network's ability to capture meaningful geometric relationships. To improve correspondence modeling, subsequent methods have integrated optical flow to provide coarse alignment between frames. While this alleviates some misalignments, the aggregation is still uniform: pixels are shifted according to flow but combined without regard to depth discontinuities or object boundaries, leading to blurred correspondences and unstable geometry modeling. Over long trajectories, these small misalignments accumulate into significant drift (Figure 1b) (Zhu et al., 2022; Li et al., 2021). In contrast, our approach couples optical flow with a deformable correlation mechanism. Flow provides a coarse correspondence prior, while the network learns local offsets to adaptively refine matches, enabling the capture of fine-grained inter-frame relationships beyond rigid pixel matches (Figure 1c). This design explicitly leverages both motion cues and adaptive local geometry, producing more accurate and robust alignment for visual odometry.

Our contributions are centered around tackling the two fundamental challenges in learning-based VO: the presence of outliers and the lack of explicit inter-frame geometry modeling. First, we propose FGD-VO, an end-to-end monocular VO framework that jointly optimizes feature extraction, correspondence modeling, and motion regression, eliminating the need for external geometric solvers. Second, to robustly handle unreliable correspondences,

we introduce a hybrid masking strategy that combines explicit flow-consistency cues with a learnable, content-adaptive attention mask, enabling the model to simultaneously reject gross mismatches and down-weight low-signal regions. Third, to strengthen geometric alignment, we design a flow-guided deformable (FGD) correlation module that integrates coarse optical flow guidance with learnable local offsets, capturing fine-grained inter-frame correspondences beyond rigid pixel matches. Through extensive evaluation on the KITTI odometry benchmark, we demonstrate that these components jointly yield state-of-the-art accuracy and robustness across diverse driving scenarios.

## 2 Related Work

Visual odometry (VO) aims to estimate the 6-DoF camera motion from image sequences, and deep learning has brought substantial advances in this field. Existing learning-based VO methods can be broadly categorized into supervised and self-supervised approaches. Supervised methods, such as PoseNet(Kendall et al., 2015) and DeepVO(Wang et al., 2017), utilize convolutional and recurrent neural networks to directly regress camera poses from monocular RGB sequences, demonstrating the capability of neural networks to implicitly learn spatial and temporal features for motion estimation.

In parallel, self-supervised methods have emerged to leverage unlabeled video sequences for training. Representative work, such as SfM-Learner(Bian et al., 2019), jointly optimizes ego-motion, depth, and optical flow under a photometric consistency constraint. Extensions include UnDeepVO(Li et al., 2018), which incorporates stereo geometric constraints to recover metric scale, and DF-VO(Zhan et al., 2021), which introduces differentiable pose graph optimization to enforce temporal consistency and reduce long-term drift.

Recent research has increasingly explored the use of transformer architectures in visual odometry, motivated by their strong capability to model long-range dependencies and capture global contextual information. By leveraging self-attention mechanisms, transformers can jointly reason about spatial and temporal relationships in image sequences, offering advantages over traditional recurrent networks in handling large viewpoint changes, motion blur, and complex scene dynamics. Representative works, such as TartanVO(Wang et al., 2021b), TSformer-VO(Françani & Maximo, 2025), SWformer-VO(Wu & Zhu, 2024), adopt a transformer-based encoder–decoder structure to enhance feature aggregation across frames, leading to more robust and accurate pose estimation. However, Transformer-based methods remain predominantly appearance-driven: self-attention captures correlations in feature space but does not explicitly enforce pixel-level geometric alignment. As a result, they may struggle in scenarios with repetitive textures, large parallax, or occlusions, where geometry cues are indispensable.

## 3 Methodology

### 3.1 Overall Pipeline

The overall structure of our proposed FGD-VO is shown in Figure 2. Given two successive monocular RGB frames $I_{t-1}, I_t \in \mathbb{R}^{H \times W \times 3}$, our objective is to estimate the inter-frame camera motion $\Delta \mathbf{T} \in \mathrm{SE}(3)$, parameterized by a rotation quaternion $\mathbf{q} \in \mathbb{R}^4$ and a translation vector $\mathbf{t} \in \mathbb{R}^3$.

In detail, we first input the consecutive monocular images into the hierarchical feature extraction module described in Section 3.2, where an FPN backbone is employed to obtain multi-scale image representations. In parallel, optical flow is estimated and downsampled to match each feature resolution, producing flow maps for every scale. Next, Section 3.3 introduces our Flow-Guided Deformable (FGD) feature fusion module, which integrates coarse correspondence from flow with learnable offsets for precise alignment. Section 3.4 presents the hybrid masking strategy designed to filter unreliable matches and emphasize valid regions. Finally, as detailed in Section 3.5, the fused features are used for iterative pose regression, progressively refining the motion estimates from coarse to fine.

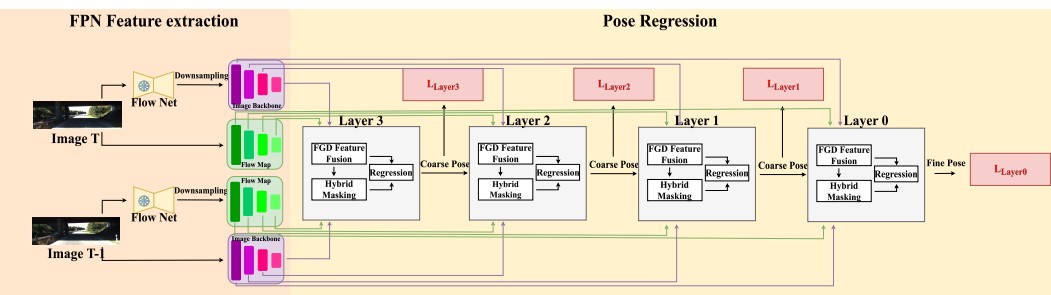

Figure 2: The pipeline of our proposed Flow-Guided Deformable Visual Odometry (FGD-VO). Our architecture consists of an FPN-based image backbone for multi-scale feature extraction, a flow network for coarse correspondence estimation, and a flow-guided deformable (FGD) feature fusion module with hybrid masking for outlier suppression. The pose is regressed iteratively from coarse to fine scales, progressively refining the estimation through fused features in shallower layers.

### 3.2 FPN Feature Extraction

We first feed two consecutive monocular images $I_{T-1}, I_T \in \mathbb{R}^{H \times W \times 3}$ into HRNet (Wang et al., 2020) backbones to extract their respective visual features. This results in two sets of multi-scale feature maps:

$$\{F_{T-1}^l, F_T^l \in \mathbb{R}^{C_l \times H_l \times W_l} \mid l = 3, 2, 1, 0\}. \tag{1}$$

where $l$ indexes the pyramid level, $C_l$ denotes the number of channels, and $(H_l, W_l)$ are the spatial resolutions, which progressively decrease as $l$ increases.

In parallel, we employ a RAFT (Teed & Deng, 2020) network to estimate both forward and backward dense optical flow fields between $I_{T-1}$ and $I_T$. Specifically, the forward flow $\Phi^{fw} \in \mathbb{R}^{2 \times H \times W}$ captures the pixel displacement from $I_{T-1}$ to $I_T$, while the backward flow $\Phi^{bw} \in \mathbb{R}^{2 \times H \times W}$ describes the reverse motion from $I_T$ to $I_{T-1}$. Both full-resolution flow maps are downsampled to match each feature scale, resulting in

$$\{\Phi^{fw,l}, \Phi^{bw,l} \in \mathbb{R}^{2 \times H_l \times W_l} \mid l = 3, 2, 1, 0\}. \tag{2}$$

At each pyramid level $l$, we thus obtain a tuple $\left(F^l, \Phi^{fw,l}, \Phi^{bw,l}\right)$, where $F^l$ denotes the HRNet-derived image features and $\Phi^{fw,l}$ / $\Phi^{bw,l}$ represent the corresponding forward and backward RAFT-derived flow maps. These multi-scale representations serve as inputs to the subsequent fusion and masking stage.

### 3.3 Flow-Guided Deformable Feature Fusion

A key limitation of prior VO frameworks is that they either rely on fixed pixel-to-pixel correspondences (e.g., concatenation-based methods) or adopt global flow warping without adaptivity. To address this gap, we introduce the Flow-Guided Deformable (FGD) Feature Fusion module, which explicitly couples optical flow priors with learnable deformable offsets. Unlike existing designs that treat flow alignment or local offsets in isolation, our module integrates them hierarchically: (i) a global stage that leverages flow as a coarse motion prior but augments it with learned offsets to handle large displacements, and (ii) a local stage that further refines alignment at sub-pixel resolution. This two-level design allows the network to move beyond rigid flow warping, capturing both wide-baseline motion and fine-grained geometric variations (Figure 3).

Global-Level Fusion. At pyramid level $l$, we concatenate the source features $F_{T-1}^l$ and target features $F_T^l$, and process them with a learnable offset encoder $\mathcal{E}_G$ to predict a global offset map $O_G^l$:

$$O_G^l = \mathcal{E}_G([F_{T-1}^l, F_T^l]). \tag{3}$$

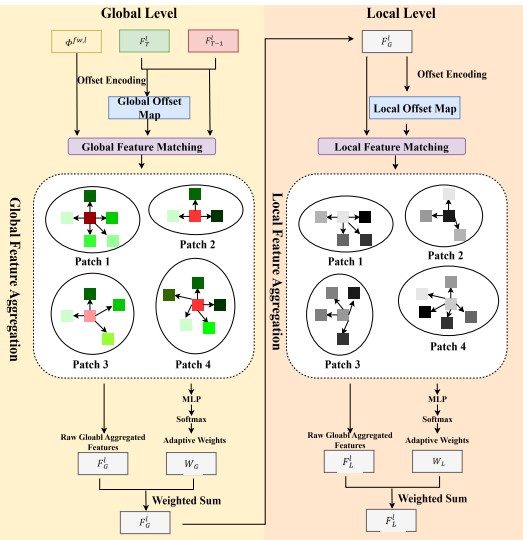

Figure 3: Architecture of the proposed Flow-Guided Deformable Feature Fusion (FGD) module. The fusion process is conducted in two complementary stages. At the global level (left), forward optical flow and learnable global offsets are combined to refine coarse correspondences between the source and target features, enabling global patch-wise aggregation. At the local level (right), the output of the global stage is further refined using learnable local offsets to capture fine grained feature alignment within each patch. Adaptive weights predicted by MLPs are applied in both stages to aggregate matched features.

Here, optical flow $\Phi^{fw,l}(p)$ serves as a coarse but informative prior, locating the approximate correspondence for each source position $p$. Instead of treating this alignment as fixed, we refine it by adding learnable offsets $O_G^l(p, q)$, which adaptively shift the sampling points. The global feature matching is thus:

$$F_G^l(p) = \sum_{q \in N_G} W_G^l(p, q) \cdot F_T^l\big(p + \Phi^{fw,l}(p) + O_G^l(p, q)\big), \tag{4}$$

where $W_G^l(p, q)$ are softmax-normalized adaptive weights. This stage enables the model to explicitly handle large inter-frame displacements, going beyond rigid flow warping.

Local-Level Fusion. While global offsets are effective for coarse alignment, they may still overlook subtle geometric cues around object edges or in texture-sparse regions. To address this, we introduce a complementary local refinement stage. The aggregated features $F_G^l$ are passed to a local offset encoder:

$$O_L^l = \mathcal{E}_L(F_G^l). \tag{5}$$

For each reference position $p$, a set of $N_L$ nearby sampling points are adaptively shifted by $O_L^l(p, q)$, yielding the refined representation:

$$F_L^l(p) = \sum_{q \in N_L} W_L^l(p, q) \cdot F_G^l\big(p + O_L^l(p, q)\big), \tag{6}$$

where $W_L^l(p, q)$ are learned adaptive weights. This refinement complements the global stage by capturing sub-pixel alignments and fine-grained structural variations, which are essential for robust pose estimation.

3.4 Hybrid Masking

A central challenge in visual odometry is to reliably suppress unreliable correspondences while preserving informative regions. Purely explicit masks, such as those derived from flow or depth consistency, are effective at rejecting gross mismatches but often discard

valid pixels near occlusions or complex motion boundaries. Conversely, purely implicit masks learned by the network offer flexibility, yet lack the geometric grounding to robustly filter dynamic outliers. To overcome these complementary weaknesses, we propose a hybrid masking strategy that couples explicit forward–backward flow consistency with an implicitly learned attention prior refined hierarchically across scales. This design allows our network to retain the geometric reliability of explicit masks while adaptively emphasizing informative regions through implicit learning.

**Hierarchical implicit mask.** Let $A^{l+1} \in \mathbb{R}^{C'_l \times H_{l+1} \times W_{l+1}}$ denote the attention map from the coarser level. We upsample and refine it with current features to form a cross-scale prior:

$$\bar{A}^{l+1} = \mathcal{U}(A^{l+1}; H_l, W_l), \qquad R^l = \mathcal{G}([\bar{A}^{l+1}, F_T^l]), \tag{7}$$

where $\mathcal{U}$ is bilinear upsampling and $\mathcal{G}$ is a lightweight conv/MLP block. We then construct the mask logits by combining the refined prior, the fused features, and the current image features:

$$L^l = \mathcal{H}([F_{T-1}^l, F_T^l, R^l, \Psi(F_L^l)]) \in \mathbb{R}^{C'_l \times H_l \times W_l}, \tag{8}$$

with $\Psi$ and $\mathcal{H}$ implemented as $1{\times}1$ conv/MLP towers. This hierarchical refinement encourages spatial consistency across scales while allowing flexible adaptation to local scene content.

**Explicit flow-consistency mask.** To provide a geometry-grounded reliability signal, we compute a forward–backward flow consistency check. Given forward flow $\Phi^{fw,l}$ (from $I_{T-1}$ to $I_T$) and backward flow $\Phi^{bw,l}$ (from $I_T$ to $I_{T-1}$), we measure their consistency at each pixel $p = (x, y)$:

$$p' = p + \Phi^{fw,l}(p), \tag{9}$$

$$\Phi^{bw,l\to}(p) = \mathcal{W}(\Phi^{bw,l}, p'), \tag{10}$$

$$D^l(p) = \left\| \Phi^{fw,l}(p) + \Phi^{bw,l\to}(p) \right\|_2, \tag{11}$$

where $\mathcal{W}(\cdot)$ denotes bilinear warping at the forward-projected location $p'$. Out-of-bounds correspondences are masked, and min–max normalization is applied to $D^l(p)$ to yield the explicit confidence:

$$M_{\text{flow}}^l(p) = 1 - \text{Norm}(D^l(p)). \tag{12}$$

This confidence map highlights pixels whose forward–backward pairs are mutually consistent, while suppressing those degraded by occlusion or unreliable motion.

**Hybridization and spatial softmax.** Finally, we combine the implicit logits with the explicit confidence. The logits $L^l$ are modulated by $M_{\text{flow}}^l$ via element-wise multiplication, and a positional embedding $E^l \in \mathbb{R}^{1 \times H_l W_l}$ is added to maintain spatial awareness:

$$\widetilde{L}_c^l(p) = L_c^l(p) \cdot M_{\text{flow}}^l(p) + E^l(p). \tag{13}$$

A per-channel spatial softmax is then applied:

$$A_c^l(p) = \frac{\exp(\widetilde{L}_c^l(p))}{\sum_{s \in \Omega_l} \exp(\widetilde{L}_c^l(s))}, \quad \sum_{p \in \Omega_l} A_c^l(p) = 1, \tag{14}$$

where $\Omega_l$ is the set of spatial locations at level $l$.

### 3.5 Iterative Pose Refinement

Inspired by (Wang et al., 2021a; Liu et al., 2024), we adopt a coarse-to-fine refinement strategy to progressively improve the relative pose between two consecutive frames. We first obtain the initial quaternion $q^L \in \mathbb{R}^4$ and translation $t^L \in \mathbb{R}^3$ from the coarsest level $L$. At each finer level $l$ ($l < L$), the network predicts residual updates $(\Delta q^l, \Delta t^l)$ based on the features at that scale, and refines the pose as:

$$q^l = \Delta q^l \otimes q^{l+1}, \tag{15}$$

$$t^l = t^{l+1} + \mathcal{R}(q^{l+1}) \cdot \Delta t^l, \tag{16}$$

Table 1: Translational error $t_{\mathrm{rel}}$ (%) and rotational error $r_{\mathrm{rel}}$ (deg/100m) on KITTI odometryGeiger et al. (2012) for sequences 03, 04, 05, 06, 07, 10. Blue numbers indicate the second-best results, and red denotes the best results. Methods marked with an asterisk (*) are trained on KITTI odometry sequences 00, 01, 02, 08, and 09, whereas methods without an asterisk are trained on sequences 01, 02, 08, and 09.

| Method | 03 | | 04 | | 05 | | 06 | | 07 | | 10 | | Mean | |
|---|---|---|---|---|---|---|---|---|---|---|---|---|---|---|
| | $t_{\mathrm{rel}}$ | $r_{\mathrm{rel}}$ | $t_{\mathrm{rel}}$ | $r_{\mathrm{rel}}$ | $t_{\mathrm{rel}}$ | $r_{\mathrm{rel}}$ | $t_{\mathrm{rel}}$ | $r_{\mathrm{rel}}$ | $t_{\mathrm{rel}}$ | $r_{\mathrm{rel}}$ | $t_{\mathrm{rel}}$ | $r_{\mathrm{rel}}$ | $t_{\mathrm{rel}}$ | $r_{\mathrm{rel}}$ |
| DeepVO(Wang et al., 2017)* | 8.49 | 6.89 | 7.19 | 6.97 | 2.62 | 3.61 | 5.42 | 5.82 | 3.91 | 4.60 | 8.11 | 8.83 | 5.96 | 6.12 |
| Saputra et al. (2019)* | 8.12 | 3.47 | 7.57 | 2.61 | 5.77 | 2.00 | 7.66 | 1.66 | 6.79 | 3.00 | 8.29 | 2.94 | 7.37 | 2.67 |
| ESPVO(Wang et al., 2018)* | 6.72 | 6.46 | 6.33 | 6.08 | 3.35 | 4.93 | 7.24 | 7.29 | 3.52 | 5.02 | 9.77 | 10.20 | 6.15 | 6.66 |
| Wu & Zhu (2024) | 11.66 | 6.73 | 5.66 | 1.60 | 11.45 | 4.50 | 10.31 | 2.92 | 11.35 | 7.71 | 9.27 | 3.04 | 9.95 | 9.04 |
| Françani & Maximo (2025) | 14.73 | 6.99 | 8.24 | 4.85 | 9.62 | 3.63 | 25.05 | 8.44 | 17.01 | 6.36 | 15.46 | 4.67 | 15.01 | 5.82 |
| Ours* | 3.87 | 1.93 | 2.87 | 1.67 | 4.59 | 2.03 | 5.61 | 1.83 | 5.36 | 2.99 | 4.61 | 1.97 | 4.48 | 2.07 |

Table 2: Translational error $t_{\mathrm{rel}}$ (%) and rotational error $r_{\mathrm{rel}}$ (deg/100m) on KITTI odometry(Geiger et al., 2012) for sequences 09 and 10. All methods, except for JPerceiver(Zhao et al., 2022), are trained only on KITTI odometry sequences 00–08; JPerceiver(Zhao et al., 2022) is trained on sequences 00–06 and 08–09; "–" means not reported.

| Method | 09 | | 10 | | Mean | |
|---|---|---|---|---|---|---|
| | $t_{\mathrm{rel}}$ | $r_{\mathrm{rel}}$ | $t_{\mathrm{rel}}$ | $r_{\mathrm{rel}}$ | $t_{\mathrm{rel}}$ | $r_{\mathrm{rel}}$ |
| SfMLearner(Zhou et al., 2017) | 11.12 | 4.10 | 12.98 | 4.26 | 12.05 | 4.18 |
| Wang et al. (2019) | 9.30 | 3.50 | 7.21 | 3.90 | 8.26 | 3.70 |
| MonoDepth2(Godard et al., 2019) | 11.75 | 3.29 | 7.98 | 3.36 | 9.87 | 3.33 |
| Zhan et al. (2018) | 11.92 | 3.60 | 12.62 | 3.43 | 12.27 | 3.52 |
| SC-SfMLearer(Bian et al., 2019) | 11.20 | 3.35 | 10.10 | 4.96 | 10.65 | 4.16 |
| Geonet(Yin & Shi, 2018) | 27.50 | 9.80 | 25.62 | 9.14 | 26.56 | 9.47 |
| Li et al. (2019b) | 8.10 | 2.81 | 12.90 | 3.17 | 10.50 | 2.99 |
| Masked GANs(Zhao et al., 2020) | 8.71 | 3.10 | 9.63 | 3.42 | 9.17 | 3.26 |
| JPerceiver(Zhao et al., 2022) | – | – | 7.52 | 3.83 | 7.52 | 3.83 |
| Zou et al. (2020) | 3.49 | 1.00 | 5.81 | 1.80 | 4.65 | 1.40 |
| Ours | 2.55 | 0.97 | 4.20 | 1.50 | 3.38 | 1.24 |

where $\otimes$ denotes quaternion multiplication, and $\mathcal{R}(q)$ converts a quaternion into the corresponding rotation matrix. This hierarchical process incrementally corrects both rotation and translation, enabling more accurate alignment at higher resolutions.

### 3.6 Loss Function

Following Li et al. (2019a), at each refinement stage $l$, the network outputs an estimated quaternion $q^l$ and translation $t^l$. The loss at this stage supervises both components, with separate learnable scaling factors for balancing rotation and translation terms:

$$\mathcal{L}^l = \|t_{\mathrm{gt}} - t^l\|_1 \cdot e^{-s_t} + s_t + \|q_{\mathrm{gt}} - q^l\|_2 \cdot e^{-s_q} + s_q, \tag{17}$$

where $(t_{\mathrm{gt}}, q_{\mathrm{gt}})$ are the ground-truth translation and quaternion, $s_t$ and $s_q$ are trainable parameters that adaptively weight the two terms, and $\|\cdot\|_1$, $\|\cdot\|_2$ denote the $\ell_1$ and $\ell_2$ norms, respectively.

The overall objective aggregates losses from all levels, with layer-specific weights $\beta^l$:

$$\mathcal{L} = \sum_{l=1}^{L} \beta^l \, \mathcal{L}^l, \tag{18}$$

where $L$ is the number of refinement levels. This multi-scale supervision enforces consistent pose estimation quality across the coarse-to-fine hierarchy.

## 4 Experiment

### 4.1 Experimental Results

We evaluate the proposed method on the KITTI odometry benchmark (Geiger et al., 2012) using the standard evaluation protocol. The evaluation metrics are the average translational error $t_{\mathrm{rel}}$ (in %), the average rotational error $r_{\mathrm{rel}}$ (in deg/100 m), and the absolute trajectory

Table 3: Absolute Trajectory Error (ATE, m) on KITTI odometry sequences 07–10. Our method is trained on sequences 00–06, while the compared hybrid VO and traditional VO approaches do not require training. The reported metric is the average ATE for each sequence, with blue numbers indicating the second-best results and red denoting the best results.

| Method | 07 | 08 | 09 | 10 | Mean |
|---|---|---|---|---|---|
| ORB-SLAM2(Mur-Artal & Tardós, 2017) | 4.56 | 35.67 | 79.45 | 7.89 | 31.89 |
| ORB-SLAM3(Campos et al., 2021) | 8.88 | 60.89 | 77.20 | 9.85 | 39.20 |
| DPVO(Teed et al., 2023) | 17.55 | 119.60 | 77.20 | 12.55 | 56.73 |
| DPV-SLAM++(Lipson et al., 2024) | 1.52 | 110.90 | 76.70 | 13.70 | 50.71 |
| MambaVO++(Wang et al., 2025) | 1.70 | 105.42 | 63.24 | 10.51 | 45.21 |
| DROID-VO(Teed & Deng, 2021) | 24.20 | 64.55 | 71.80 | 16.91 | 44.36 |
| Ours | 11.19 | 24.94 | 36.51 | 8.47 | 20.45 |

error ATE (m). Results are reported for each sequence individually and averaged over the selected sequences, following established practice. To ensure a fair comparison with existing approaches, we follow multiple commonly used training/testing splits reported in prior works (Wang et al., 2017; Li et al., 2018). For each split, the model is trained and evaluated independently, allowing us to assess performance under different data partition settings and avoid bias introduced by variations in dataset division.

**Comparison with Learning-based Odometry.** We first compare our approach with representative learning-based visual odometry methods, which include classical supervised approaches such as DeepVO (Wang et al., 2017), unsupervised approaches such as SfM-Learner (Zhou et al., 2017), as well as recent multi-task frameworks like JPerceiver (Zhao et al., 2022) that jointly integrate perception and pose estimation, the results are reported in Table 1 and Table 2. Across most sequences, our method consistently achieves the lowest translational error $t_{rel}$ and rotational error $r_{rel}$, highlighting its advantage over both traditional supervised/unsupervised paradigms and more advanced multi-task designs. For example, compared with DeepVO(Wang et al., 2017) and ESPVO(Wang et al., 2018), our model reduces the mean translational error by more than 25% and significantly improves rotational accuracy. Even when compared with recent transformer-based odometry frameworks, such as SWformerVO(Wu & Zhu, 2024) and TSformerVO(Françani & Maximo, 2025), our approach still yields a clear performance margin, especially on long trajectories where feature alignment becomes challenging. These results demonstrate that our hybrid masking design and flow guided strategy allow the model to surpass purely learning-based odometry approaches, highlighting the robustness of our method under diverse motion patterns.

**Comparison with Hybrid and Traditional Visual Odometry.** We further compare our method with both hybrid VO approaches (e.g., MambaVO++ (Wang et al., 2025), DPV-SLAM++ (Lipson et al., 2024)) and traditional geometry-based methods (e.g., ORB-SLAM2 (Mur-Artal & Tardós, 2017), ORB-SLAM3 (Campos et al., 2021)). As shown in Table 3, our model consistently achieves superior results across test sequences, 07–10. Notably, our approach reduces the mean ATE to 20.45 m, representing a substantial improvement over both categories of methods. In particular, compared to geometry-based ORB-SLAM2 and ORB-SLAM3, which are highly sensitive to challenging visual conditions, our approach shows significant robustness on sequences 09 and 10. Compared to hybrid VO methods, which integrate learning-based components with geometric constraints, our method still delivers lower errors.

## 4.2 Ablation Study

To investigate the contribution of each component in our framework, we perform ablation experiments by removing or modifying individual modules and comparing the results on the KITTI odometry benchmark. Specifically, we adopt sequences 00–06 for training and 07–10 for testing.

Table 4: Ablation on KITTI odometry sequences 00–10. We group variants into three categories: (a) Hybrid Masking (explicit/implicit removal and no mask), (b) FGD (removing flow guidance or deformable correlation), and (c) the Full model. Each cell reports translational error $t_{\rm rel}$ (%) and rotational error $r_{\rm rel}$ (deg/100m). Red denotes the best results; "*" denotes the training sets.

| Category | Method | 00* | | 01* | | 02* | | 03* | | 04* | | 05* | | 06* | | 07 | | 08 | | 09 | | 10 | | Mean | |
|---|---|---|---|---|---|---|---|---|---|---|---|---|---|---|---|---|---|---|---|---|---|---|---|---|---|
| | | $t_{\rm rel}$ | $r_{\rm rel}$ | $t_{\rm rel}$ | $r_{\rm rel}$ | $t_{\rm rel}$ | $r_{\rm rel}$ | $t_{\rm rel}$ | $r_{\rm rel}$ | $t_{\rm rel}$ | $r_{\rm rel}$ | $t_{\rm rel}$ | $r_{\rm rel}$ | $t_{\rm rel}$ | $r_{\rm rel}$ | $t_{\rm rel}$ | $r_{\rm rel}$ | $t_{\rm rel}$ | $r_{\rm rel}$ | $t_{\rm rel}$ | $r_{\rm rel}$ | $t_{\rm rel}$ | $r_{\rm rel}$ | $t_{\rm rel}$ | $r_{\rm rel}$ |
| (a)Masking | Ours w/o explicit mask | 27.59 | 1.56 | 2.19 | 1.61 | 4.52 | 1.69 | 3.60 | 1.79 | 3.48 | 2.41 | 3.90 | 1.90 | 4.22 | 2.11 | 5.20 | 3.38 | 7.86 | 3.32 | 6.82 | 2.75 | 9.62 | 3.74 | 7.18 | 2.38 |
| | Ours w/o implicit mask | 6.33 | 2.49 | 3.93 | 1.03 | 5.63 | 1.98 | 3.63 | 2.16 | 2.26 | 0.86 | 5.97 | 2.24 | 7.11 | 2.55 | 8.30 | 4.79 | 6.81 | 2.44 | 4.61 | 1.75 | 4.10 | 2.12 | 5.33 | 2.21 |
| | Ours w/o mask | 7.91 | 3.62 | 4.05 | 1.25 | 8.14 | 3.04 | 6.57 | 4.66 | 3.15 | 2.16 | 4.67 | 2.30 | 9.76 | 3.91 | 4.52 | 2.79 | 5.30 | 2.31 | 3.90 | 1.91 | 4.78 | 3.36 | 5.70 | 2.84 |
| (b) FGD | Ours w/o flow guidance | 18.53 | 8.56 | 9.30 | 2.64 | 10.47 | 4.78 | 26.96 | 9.81 | 3.07 | 3.46 | 13.57 | 6.57 | 4.37 | 2.69 | 24.99 | 16.71 | 25.01 | 11.36 | 22.66 | 9.16 | 22.57 | 10.44 | 16.77 | 7.83 |
| | Ours w/o deformable | 5.64 | 2.79 | 5.53 | 1.63 | 4.61 | 2.14 | 4.90 | 3.52 | 1.56 | 1.62 | 3.42 | 2.31 | 2.84 | 1.67 | 4.62 | 4.16 | 7.40 | 3.35 | 4.72 | 2.31 | 5.69 | 3.29 | 4.63 | 2.61 |
| (c) Full | Ours (full) | 3.51 | 1.52 | 5.21 | 1.50 | 3.18 | 1.17 | 2.56 | 1.32 | 2.42 | 1.36 | 2.73 | 1.13 | 2.30 | 1.22 | 5.43 | 2.66 | 4.57 | 1.91 | 4.61 | 2.03 | 4.95 | 2.26 | 3.77 | 1.64 |

**Effect of Hybrid Masking.** We assess the contribution of the proposed hybrid masking strategy—which combines an explicit flow-consistency mask with an implicitly learned mask refined across scales—by disabling each component in turn; the results are summarized in Table 4(a). Removing the explicit mask ("w/o explicit") leads to a noticeable increase of both errors across most sequences (e.g., mean $t_{\rm rel}$ rises from 3.77 to 7.18 and mean $r_{\rm rel}$ from 1.64 to 2.38), indicating that forward–backward flow consistency is effective at gating unreliable correspondences arising from dynamic objects and occlusions. Eliminating the implicit mask ("w/o implicit") also degrades performance (mean $t_{\rm rel}$ 5.33, $r_{\rm rel}$ 2.21); without the learnable attention, the network struggles to down-weight low-signal regions that still pass the explicit check and to emphasize geometry-informative pixels. When both masks are removed ("w/o mask"), the drop is the largest (mean $t_{\rm rel}$ 5.70, $r_{\rm rel}$ 2.84), effectively reducing the fusion to uniform pooling and making the model vulnerable to outliers. Together, these observations suggest complementary roles: the explicit mask reliably screens out gross mismatches by flow consistency, while the implicit mask performs fine-grained, data-driven selection that adapts to scene content and scale.

**Effect of Flow-Guided Deformable Module.** To examine the role of the Flow-Guided Deformable module, we consider two ablated variants by removing its major components. The first variant ("Ours w/o flow guidance") discards the flow-based guidance, which deprives the model of explicit motion priors and results in noticeably higher translational and rotational errors across sequences. The second variant ("Ours w/o deformable") replaces the learnable deformable sampling with a naive strategy that directly aggregates features from the nearest $N$ points in a fixed neighborhood. This design removes the model's ability to adaptively select informative positions for fusion, thereby limiting its flexibility in handling diverse geometric structures. As shown in Table 4(b), both modifications lead to clear performance degradation, whereas the complete Flow-Guided Deformable module achieves the most accurate odometry estimates. These results demonstrate that flow guidance provides valuable motion information, while deformable correlation improves feature alignment flexibility, and that their joint contribution is crucial for robust pose estimation.

## 5 Conclusion

In this work, we presented a novel visual odometry framework that integrates hybrid masking and a flow-guided deformable module to enhance pose estimation accuracy. Through extensive evaluations on the KITTI odometry benchmark, our method consistently outperformed both traditional and learning-based approaches, achieving lower translational and rotational errors across diverse sequences. Ablation studies further verified the effectiveness of each proposed component, demonstrating that hybrid masking improves robustness to noisy correspondences, while the flow-guided deformable module strengthens feature alignment and adaptability. Despite being trained on limited sequences, our framework still surpasses competing methods trained on larger datasets, highlighting its efficiency and strong generalization capability. These results underscore the potential of our design to advance reliable and accurate odometry in real-world autonomous driving scenarios.

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

# A Appendix

## A.1 Implementation Details

We conduct experiments on the KITTI odometry benchmark (Geiger et al., 2012), using only the monocular left camera images as input. Since ground-truth trajectories are available exclusively for sequences 00–10, our training and evaluation are restricted to this subset. This ensures that all reported results are obtained with accurate pose annotations, allowing for a fair comparison with existing visual odometry methods.

The input images are cropped and resized to a resolution of $192 \times 640$ to maintain consistency across the dataset. An HRNet-W18 backbone, pretrained on ImageNet, is employed for feature extraction, serving as the foundation of our model pipeline. All experiments are conducted using PyTorch 1.10.1 on an NVIDIA RTX A6000 Ada GPU. For optimization, we adopt the Adam optimizer with parameters $\beta_1 = 0.9$ and $\beta_2 = 0.999$, and the initial learning rate is set to 0.001. The learning rate follows an exponential decay schedule with a decay step of 200,000 and a minimum learning rate of 0.00001. To improve generalization, we also apply a dropout regularization with a rate of 0.25. The batch size is fixed to 4 during training.

Furthermore, we assign weighting coefficients $\alpha^l$ to the outputs of different refinement layers ($l = 0, 1, 2, 3$), with values set to 0.2, 0.4, 0.8, and 1.6, respectively. Initial values of the learnable loss-balancing parameters $k_x$ and $k_q$ are set to 0.0 and $-2.5$. We follow the widely adopted KITTI evaluation protocol, reporting two key metrics: (1) average sequence translational RMSE drift $t_{\text{rel}}$ (%); (2) average sequence rotational RMSE drift $r_{\text{rel}}$ (deg/100m); (3) absolute trajectory error ATE(m).

### A.1.1 Main model

The proposed network follows a hierarchical multi-scale design, where image features are progressively extracted and refined through multiple stages before being aggregated for final pose estimation. The backbone of the model is a Pyramid HRNet-W32 (Wang et al., 2020), pretrained on ImageNet, which generates multi-resolution feature maps at four pyramid levels. The input images are resized to $192 \times 640$ to ensure consistency across the KITTI odometry dataset. At each pyramid level ($l = 0, 1, 2, 3$), features from consecutive frames are processed jointly with optical flow guidance to facilitate motion-aware feature alignment. Optical flow is estimated using a pretrained RAFT (Teed & Deng, 2020) network, and downsampled into four pyramid resolutions via bilinear interpolation, ensuring that motion information is available at all feature scales.

Layer-wise architecture. At each pyramid level, Flow-Guided Attention (FGA) modules are employed to fuse features from the two input frames using the corresponding optical flow. The fused representations are concatenated with backbone features and passed through Multi-Layer Perceptrons (MLPs) implemented as $1 \times 1$ convolutions, batch normalization, and ReLU activations. Each level employs a pair of separate MLP branches to predict feature embeddings and attention weights, which are further combined with flow consistency masks to suppress unreliable regions. This design enables the model to robustly filter out mismatched correspondences and focus on geometrically consistent regions. Positional embeddings are also added at each level to preserve spatial information in the flattened feature tokens.

Pose regression. Each pyramid level outputs two branches: one for rotation (quaternion representation) and one for translation (3D vector). To stabilize training, dropout regularization with a rate of 0.25 is applied separately to the translation and rotation branches. The outputs of deeper layers are progressively refined by combining the residual predictions with estimates from coarser levels. Specifically, rotations are updated through quaternion multiplication, while translations are transformed by quaternion-based operations, ensuring consistency across levels. This hierarchical refinement continues from the coarsest feature map (layer 3) to the finest feature map (layer 0), where the final pose estimates are obtained.

Parameter configuration. The channel dimensions of the backbone features are $(64, 128, 256, 512)$ for layers 0 to 3, respectively. Each FGA module uses a local MLP with hidden size $(64, 64)$. The refinement MLPs for each level are configured as follows: layer 3 uses $(128, 64)$, layer 2 uses $(128, 64)$, layer 1 uses $(128, 64)$, and layer 0 uses $(128, 64)$. The up-projection layers map the outputs of these MLPs to 256-dimensional embeddings for further processing. The quaternion and translation heads at each level are implemented as $1 \times 1$ convolutions that reduce the feature embeddings to 4-dimensional and 3-dimensional vectors, respectively. Initial positional embeddings are randomly initialized with truncated normal distribution ($\sigma = 0.02$).

Pipeline summary. Overall, the pipeline can be summarized as follows: (1) Extract multi-scale features from consecutive frames using HRNet-W32. (2) Compute dense optical flow between frames using RAFT, and downsample it into four pyramid levels. (3) Fuse features and flow information at each level using Flow-Guided Attention. (4) Apply flow-consistency masking and positional embedding to suppress unreliable regions. (5) Pass the fused features through level-specific MLPs to obtain translation and rotation predictions. (6) Hierarchically refine the pose from coarse to fine levels using quaternion multiplication and residual translation updates. This hierarchical design leverages both appearance and motion cues, enabling the model to achieve robust pose estimation in challenging visual odometry scenarios.

A.1.2 Flow-Guided Deformable (FGD) Module

Given two pyramid features from consecutive images, $F_{T-1}^l, F_T^l \in \mathbb{R}^{C \times H_l \times W_l}$, and the forward optical flow $\Phi^{fw,l} \in \mathbb{R}^{2 \times H_l \times W_l}$, the FGD module aligns and fuses features in two successive stages: a flow-guided global aggregation that searches a coarse neighborhood along the flow direction, followed by a deformable local aggregation that refines correspondences with sub-pixel precision.

Coordinate preparation. Let $G \in \mathbb{R}^{H_l \times W_l \times 2}$ be the pixel coordinate grid $G(x, y) = (x, y)$. The forward projection of each location $p$ is

$$T(p) = G(p) + \Phi^{fw,l}(p) \in \mathbb{R}^2. \tag{19}$$

All sampling operations use differentiable bilinear warping via grid_sample with align_corners=True and padding_mode=border.

(1) Flow-guided global aggregation. We first predict $K$ coarse offsets around the flow-projected target using the concatenated features:

$$O_G = \text{Conv}_{3 \times 3}\big([F_{T-1}^l, F_T^l]\big) \in \mathbb{R}^{H_l \times W_l \times K \times 2}, \tag{20}$$

$$S_G(p, k) = T(p) + O_G(p, k). \tag{21}$$

Features from $F_T^l$ are sampled at $S_G$ and paired with the source feature:

$$\widehat{F}_T^l(p, k) = \mathcal{W}\big(F_T^l, S_G(p, k)\big), \tag{22}$$

$$\Delta(p, k) = S_G(p, k) - G(p), \qquad d(p, k) = \|\Delta(p, k)\|_2, \tag{23}$$

and the global descriptor for each candidate is built by concatenating appearance and simple geometry:

$$z_G(p, k) = \big[\, F_{T-1}^l(p), \ \widehat{F}_T^l(p, k), \ \Delta(p, k), \ d(p, k) \,\big]. \tag{24}$$

Two light MLP towers process appearance and geometry, respectively, and produce an attention score over the $K$ samples:

$$\phi_G(p, k) = \text{MLP}_{\text{feat}}(z_G(p, k)), \quad \gamma_G(p, k) = \text{MLP}_{\text{geo}}\big([\Delta(p, k), d(p, k)]\big), \tag{25}$$

$$a_G(p, k) = \text{softmax}_k\Big(W_G\left[\phi_G(p, k), \gamma_G(p, k)\right]\Big). \tag{26}$$

The global fused feature is

$$Z_G(p) = \sum\nolimits_{k=1}^{K} a_G(p, k)\, \phi_G(p, k) \in \mathbb{R}^{C_G}, \tag{27}$$

where $K = \text{nsample\_coarse} = 32$ and $C_G = \text{mlp1[-1]}$ in the implementation.

(2) Deformable local refinement.   Taking $Z_G$ as a query map, we predict $k$ learnable local offsets around each reference pixel (no flow is used here) to refine alignment:

$$O_L = \text{Conv}_{3\times3}(Z_G) \in \mathbb{R}^{H_l \times W_l \times k \times 2}, \tag{28}$$

$$S_L(p,i) = G(p) + O_L(p,i), \qquad i = 1, \ldots, k. \tag{29}$$

We sample the query feature itself at these offsets and perform a second attention aggregation with geometry encoding identical to the global stage:

$$\widehat{Z}_G(p,i) = \mathcal{W}(Z_G, S_L(p,i)), \quad d_L(p,i) = \|O_L(p,i)\|_2, \tag{30}$$

$$z_L(p,i) = \left[\, Z_G(p), \ \widehat{Z}_G(p,i), \ O_L(p,i), \ d_L(p,i) \,\right], \tag{31}$$

$$\phi_L(p,i) = \text{MLP}_{\text{feat}}^{\text{loc}}(z_L(p,i)), \quad \gamma_L(p,i) = \text{MLP}_{\text{geo}}^{\text{loc}}([O_L(p,i), d_L(p,i)]), \tag{32}$$

$$a_L(p,i) = \text{softmax}_i\Big(W_L\,[\phi_L(p,i), \gamma_L(p,i)]\Big). \tag{33}$$

The final output of the FGD module at level $l$ is

$$F_{\text{FGD}}^l(p) = \sum\nolimits_{i=1}^{k} a_L(p,i)\,\phi_L(p,i) \in \mathbb{R}^{C_G}, \tag{34}$$

with $k = \text{nsample\_fine} = 9$. This output is returned in $\mathbb{R}^{C_G \times H_l \times W_l}$ after a $(H_l, W_l, C_G) \to (C_G, H_l, W_l)$ permutation.

Design rationale and hyperparameters.   The global stage (§21) searches a flow-guided, large receptive field to handle wide-baseline motion and fast dynamics; the subsequent local stage focuses on deformable sub-pixel refinement, improving alignment near boundaries and textureless regions. Geometry encoders ($1\times1$ conv with hidden size geo\_hiden=64) provide a lightweight inductive bias using coordinate differences and distances. The default channel sizes are $C = \{64, 128, 256, 512\}$ for levels $\{0, 1, 2, 3\}$, with MLP stacks mlp1 $= (128, 64, 64)$ and mlp2 $= (128, 64)$ for the global stage, and mlp1\_local $= (64, 64)$, mlp2\_local $= (64, 64)$ for the local stage. All convolutions are followed by BN and ReLU. The entire module is fully differentiable and trained end-to-end together with the pose regression heads.

Implementation notes.   Coordinates are normalized to $[-1, 1]$ before warping.   Offsets are predicted per-pixel for every spatial location, yielding tensors of shape $H_l \times W_l \times K \times 2$ (global) and $H_l \times W_l \times k \times 2$ (local). We use padding\_mode=border to ensure valid gradients near image boundaries. In practice, $K=32$ and $k=9$ strike a good balance between accuracy and speed, capturing large displacements while keeping computation tractable.

### A.1.3   Hybrid Masking

Goal.   At every pyramid level $l$, we compute a hybrid mask that combines (i) an explicit confidence from forward–backward flow consistency and (ii) an implicit attention learned from the current and coarser-level features. The hybrid mask gates the weighting branch and produces spatial weights that are used to pool the feature branch.

Explicit flow–consistency mask.   Given forward flow $\Phi^{fw,l} \in \mathbb{R}^{2 \times H_l \times W_l}$ and backward flow $\Phi^{bw,l} \in \mathbb{R}^{2 \times H_l \times W_l}$, let $G \in \mathbb{R}^{2 \times H_l \times W_l}$ be the pixel grid, $G(x,y) = (x,y)$. We first project each pixel forward

$$C^l = G + \Phi^{fw,l} \in \mathbb{R}^{2 \times H_l \times W_l}, \tag{35}$$

and bilinearly sample the backward flow at the forward-projected position (grid\_sample with align\_corners=True):

$$\Phi^{bw,l\rightarrow} = \mathcal{W}\big(\Phi^{bw,l}, C^l\big). \tag{36}$$

The forward–backward discrepancy is

$$D^l = \left\|\Phi^{fw,l} + \Phi^{bw,l\rightarrow}\right\|_2 \in \mathbb{R}^{1 \times H_l \times W_l}. \tag{37}$$

Out-of-bounds locations of $C^l$ are first marked and assigned a large value, then a per-tensor min–max normalization is applied:

$$M_{\text{flow}}^l = 1 - \text{Norm}\big(D^l\big) \in [0,1]^{1 \times H_l \times W_l}. \tag{38}$$

In implementation, invalid samples use zero padding during warping and their discrepancy is set to the maximum valid value before normalization, ensuring small confidences near boundaries.

Hierarchical implicit attention. Each level $l$ has a feature branch and a weight branch. For the feature branch, we concatenate the current image feature $F_T^l$, the FGD output at this level $F_{\text{FGD}}^l$, and a coarser prior refined at the current resolution:

$$\bar{F}^{l+1} = \text{Up}\big(F_{\text{FGD}}^{l+1}\big), \qquad U_{\text{feat}}^l = \Gamma_f\big([\bar{F}^{l+1}, F_T^l]\big), \tag{39}$$

$$X^l = [\, F_T^l, U_{\text{feat}}^l, F_{\text{FGD}}^l \,], \qquad V^l = \Phi_{\text{feat}}(X^l) \in \mathbb{R}^{C_l \times H_l \times W_l}. \tag{40}$$

For the weight branch, we construct the logits from the current feature, a coarser weight prior refined at the current scale, and the processed feature $V^l$:

$$\bar{C}^{l+1} = \text{Up}\big(C^{l+1}\big), \qquad U_{\text{w}}^l = \Gamma_w\big([\bar{C}^{l+1}, F_T^l]\big), \tag{41}$$

$$Y^l = [\, F_T^l, U_{\text{w}}^l, V^l \,], \qquad L^l = \Phi_{\text{w}}(Y^l) \in \mathbb{R}^{C_l \times H_l \times W_l}. \tag{42}$$

Here $\text{Up}(\cdot)$ is bilinear upsampling, $\Gamma_f, \Gamma_w$ are $1\times1$ convolutions that condition the coarse maps on the current features, and $\Phi_{\text{feat}}, \Phi_{\text{w}}$ are the MLP towers in code. The number of channels $C_l$ of $V^l$ and $L^l$ is the same (the last width of the MLP), which allows per-channel weighting.

Hybridization and spatial softmax. We gate the logits with the explicit confidence and add a learnable positional bias $E^l \in \mathbb{R}^{1\times1\times(H_l W_l)}$. Concretely, we flatten the spatial dimension and apply a per-channel softmax:

$$\widetilde{L}^l = \big(L^l \odot M_{\text{flow}}^l\big) \in \mathbb{R}^{C_l \times H_l \times W_l}, \tag{43}$$

$$\widehat{L}^l = \text{reshape}\big(\widetilde{L}^l\big) \;+\; E^l \;\in \mathbb{R}^{C_l \times (H_l W_l)}, \tag{44}$$

$$A_c^l(p) = \frac{\exp\big(\widehat{L}_c^l(p)\big)}{\sum_s \exp\big(\widehat{L}_c^l(s)\big)}, \quad \sum_p A_c^l(p) = 1. \tag{45}$$

The final implicit mask $A^l$ is then reshaped to $C_l \times H_l \times W_l$ and used to pool the feature branch:

$$z_c^l = \sum_p A_c^l(p)\, V_c^l(p) \;\in\; \mathbb{R}^{C_l}. \tag{46}$$

Remarks.

- The explicit mask $M_{\text{flow}}^l$ is computed at every level from the downsampled forward/backward flows; it suppresses occlusions and inconsistent motion.
- The hierarchical priors $U_{\text{feat}}^l$ and $U_{\text{w}}^l$ propagate coarse decisions to finer scales, while the positional bias $E^l$ stabilizes the spatial softmax.
- All operations are differentiable; grid_sample is used both for flow warping (explicit mask) and for feature sampling inside FGD.

## A.2 Visualization

### A.2.1 Visualization of Hybrid Masking

As illustrated in Figure A.1, both explicit and implicit weighting strategies are prone to failure in certain regions. In the orange dashed box, the explicit flow-consistency mask successfully filters out the bottom-right corner, since this region is not simultaneously visible in both input frames and thus corresponds to noise points that do not contribute to pose estimation. However, the implicit learnable mask erroneously assigns high weights to these points, misinterpreting them as informative features, which can negatively impact the model's accuracy. Conversely, in the green dashed box, the flow-based explicit mask fails to suppress several outlier points, while the implicit mask also overestimates their contribution. In both cases, the hybrid masking strategy—which integrates both flow-guided explicit cues and network-learned implicit attention—effectively eliminates these outliers and emphasizes truly informative regions. This demonstrates that our hybrid masking mechanism provides a robust balance, leveraging complementary strengths of both sources to reduce noise and improve pose estimation performance.

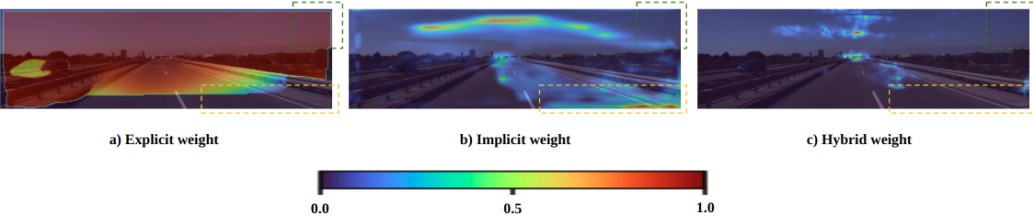

Figure A.1: Visualization of multi-source attention masks at Layer-0. The masks are up-sampled to the original input resolution (192×640) for better interpretability and overlaid on the reference image. (a) Explicit weight corresponds to the flow-consistency mask, which highlights regions with reliable optical flow. (b) Implicit weight corresponds to the learnable attention mask obtained from the network's logits via spatial softmax. (c) Hybrid weight combines both explicit and implicit cues, showing the complementary effect of flow-guided and learnable attention. The color scale ranges from blue (low weight) to red (high weight).

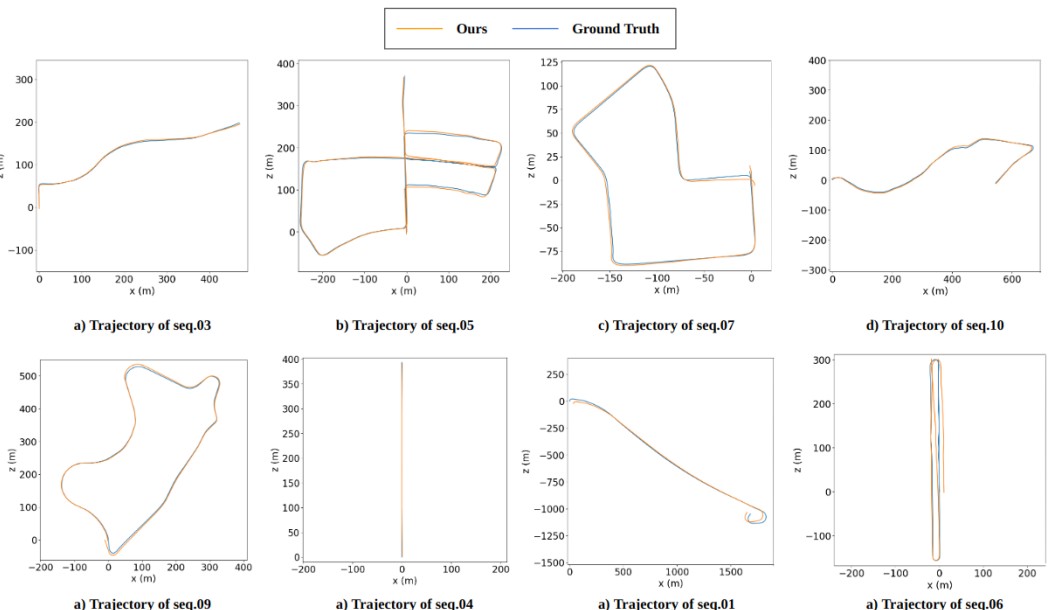

Figure A.2: Estimated trajectories generated by our method compared with the ground truth on the KITTI dataset.

### A.2.2 Visualization of Pose Estimation

As shown in Figure A.2, we provide qualitative trajectory comparisons between our method and the ground truth on multiple KITTI odometry sequences. Across diverse scenarios— including short loops (Seq.03), complex urban layouts (Seq.05), dynamic traffic scenes (Seq.07, Seq.10), and long-range drives (Seq.01)—our predictions align closely with the ground truth. These visualizations highlight that FGD-VO not only achieves low drift on short sequences but also maintains stability over long trajectories, confirming the effective-ness of our hybrid masking and flow-guided deformable correlation in suppressing outliers and preserving geometric consistency.

### A.3 Ablation Study Analysis

#### A.3.1 Masking Ablations

To isolate the contribution of each masking component, we reuse the same backbone, FGD fusion, losses, and training schedule for all variants and only change the spatial aggregation rule at every pyramid level $l$. Let $X^l \in \mathbb{R}^{C_l \times H_l \times W_l}$ be the fused feature map, $L^l \in \mathbb{R}^{C_l \times H_l \times W_l}$ the learnable mask logits produced by a $1 \times 1$ conv/MLP tower, $E^l \in \mathbb{R}^{1 \times H_l W_l}$ a positional term, and $M_{\text{flow}}^l \in \mathbb{R}^{1 \times H_l \times W_l}$ the explicit forward–backward flow consistency (Section 3.4). We form channel-wise attention $A^l \in \mathbb{R}^{C_l \times H_l \times W_l}$ and aggregate a vector

$$z_c^l = \sum_{p \in \Omega_l} A_c^l(p)\, X_c^l(p), \qquad z^l \in \mathbb{R}^{C_l},$$

where $\Omega_l$ indexes the $H_l \times W_l$ spatial sites. The three ablated settings are:

(i) No mask. Both masks are removed and we fall back to global average pooling:

$$z_c^l = \frac{1}{H_l W_l} \sum_{p \in \Omega_l} X_c^l(p). \tag{47}$$

This variant measures performance without any spatial selection.

(ii) No flow mask, learnable-only. We keep the learnable attention while dropping the flow guidance. We first build logits with a positional bias and then apply per-channel spatial softmax:

$$\widetilde{L}_c^l(p) = L_c^l(p) + E^l(p), \tag{48}$$

$$A_c^l(p) = \frac{\exp\big(\widetilde{L}_c^l(p)\big)}{\sum_{s \in \Omega_l} \exp\big(\widetilde{L}_c^l(s)\big)}. \tag{49}$$

(iii) No learnable mask, flow-only. We discard $L^l$ and derive attention solely from the explicit consistency $M_{\text{flow}}^l$. Let $D^l$ be the forward–backward discrepancy and $M_{\text{flow}}^l = 1 - \text{Norm}(D^l)$ (values in $[0,1]$). We convert it to a temperature-controlled spatial distribution, broadcast to all channels, and aggregate:

$$\alpha^l(p) = \frac{\exp\big(M_{\text{flow}}^l(p)/\tau\big)}{\sum_{s \in \Omega_l} \exp\big(M_{\text{flow}}^l(s)/\tau\big)}, \qquad A_c^l(p) = \alpha^l(p), \tag{50}$$

$$z_c^l = \sum_{p \in \Omega_l} \alpha^l(p)\, X_c^l(p). \tag{51}$$

All variants share identical backbones, FGD fusion, and training settings; only the attention formation in equation 47–equation 51 is changed. We apply the chosen rule at every scale ($l = 3, 2, 1, 0$) to ensure a fair, apples-to-apples comparison.

We further analyze the behavior of different masking strategies. An interesting observation is that removing the learnable mask accelerates the convergence speed during training, as the model no longer needs to optimize additional attention weights. However, despite this faster convergence, the final performance is inferior, indicating that the learnable mask plays a critical role in refining feature selection and improving robustness against noise. In contrast, the flow-based mask proves particularly effective in scenarios with numerous moving objects. For instance, as shown in Table 4, on sequence 00—which corresponds to a highway scene with heavy vehicle traffic—removing the explicit flow mask leads to a sharp increase in translational error $t_{\text{rel}}$, rising to 27.59%. This dramatic degradation highlights the importance of flow consistency in filtering out dynamic outliers that would otherwise mislead the odometry estimation. Overall, the learnable mask is especially beneficial in complex environments with heterogeneous textures and geometry, where static structures and subtle motion cues need to be distinguished adaptively.

### A.3.2   FGD Ablations

To further examine the role of the Flow-Guided Deformable (FGD) module, we conduct two ablation settings by selectively removing its major components. In the first case ("Ours w/o flow guidance"), we discard the optical-flow guidance and allow the network to directly learn offsets for feature sampling. Without the explicit motion prior from optical flow, the model needs to infer correspondence patterns solely from feature similarity. This forces the offset predictor to learn which patches should be fused in a purely data-driven manner, which increases learning difficulty and typically leads to noisier alignment.

In the second case ("Ours w/o deformable"), we retain the flow guidance but remove the deformable mechanism. Specifically, instead of letting the network adaptively learn sampling offsets, we directly select the nearest $N$ candidate points around the flow-warped coordinates and fuse their features. While this rigid neighborhood fusion reduces computational complexity, it also limits the flexibility of the model in handling geometric variations such as object deformation or perspective distortion. As a result, the model struggles to capture fine-grained motion cues compared to the full FGD design. Together, these ablations highlight that both flow guidance and deformable sampling are indispensable, where flow guidance provides reliable motion priors and the deformable mechanism enables adaptive feature alignment.

### A.4   AI Usage

In this work, AI tools were used exclusively for minor language polishing to improve readability and presentation. All core components of the paper—including the proposed methodology, theoretical analysis, experimental design, and result interpretation—were independently developed and written by the authors. No AI systems were involved in generating research ideas, designing algorithms, or performing the scientific contributions of this paper.

