# OpenReview forum: "FGD-VO: Flow-Guided Deformable Correlation with Hybrid Masks for Visual Odometry"
_ICLR.cc/2026/Conference — ICLR 2026 Conference Withdrawn Submission_

### Official Review · Reviewer_sGwd · 2025-10-24

**Soundness:** 2
**Presentation:** 3
**Contribution:** 1
**Rating:** 2
**Confidence:** 5

**Summary:**

This paper proposed a optical flow-guided visual odometry pipeline. A hybrid masking module is designed to filter outliers. Authors prove the effectiveness of their design on KITTI.

**Strengths:**

1.Designs are reasonable. Outliers and correspondences are truly core issue in odometry (registration) task. Authors’ design attempt to deal with these challenges.

2.Good writing. This paper has nearly no grammar issues and is very easy to follow.

**Weaknesses:**

1.The biggest concern is limited novelty. As claimed by authors, optical flow is very largely explored before previously. I reckon there may be some differences in detailed leveraging method about optical flow. However, it cannot provide enough contributions to this field. Furthermore, masking mechanism has been also widely studied, e.g, PWCLO (CVPR’21). In my opinion, this kind of weighting-based method is not novel.

2.The basic design is too simple. Both hybrid mask and flow guidance are too simple, which have limited insight.

3.The method fails to achieve state-of-the-art performance on the KITTI dataset. On Sequences 05, 06, it can be even worse than methods proposed in 2017.

4.More experiments on other datasets like nuSenes are expected to demonstrate the effectiveness.

5.Lack of time comparison. Odometry is a very efficiency-requiring task in SLAM system. Authors should evaluate the time, especially how dense flow estimations incur.

**Questions:**

Please see the weakness section. Also, I think this paper lacks theoretical contribution, which doesn't adhere to the principle of this conference.

---

### Official Review · Reviewer_5ULn · 2025-10-27

**Soundness:** 2
**Presentation:** 2
**Contribution:** 2
**Rating:** 2
**Confidence:** 4

**Summary:**

This paper presents FGD-VO, a visual odometry framework that introduces a Flow-Guided Deformable (FGD) unit to fuse optical flow cues with visual features and an attention-based masking strategy to handle dynamic and informative regions. The idea of combining explicit and implicit masks is interesting and potentially useful for motion-aware feature fusion. However, the paper has several issues related to the conceptual clarity of its modules, evaluation completeness, and manuscript organization.

**Strengths:**

1. The work attempts to jointly model explicit geometric consistency from the forward-backward flow and implicit attention learned from the data, which is reasonable and can be potentially useful in motion-aware feature fusion pipelines.
2. An extensive motivation in the introduction is clear and easy to follow.

**Weaknesses:**

1. The paper reports results only on KITTI, which is insufficient to thoroughly assess the model performance. Further evaluations on at least one more dataset with dynamic scenes (e.g., nuScenes, Argoverse, TartanAir, TUM RGB-D) are essential for a fair evaluation.

2. The paper follows inconsistent KITTI train/test splits across Tables 1-3. In particular, some tables report results of models trained on different subsets, and the training split is not specified in Tables 2-3. This makes comparison to existing work on the KITTI odometry benchmark difficult.

3. No runtime evaluation is reported, even though efficiency is crucial for visual odometry. The paper should provide runtime (ms/frame, FPS) and memory statistics for each module (RAFT, FGD fusion, and pose estimation).

4. FGD component needs further clarification and support with ablations and illustrative examples. Specifically, the module is described as combining "global and local levels" of offsets. However, according to Figure 3, this is not a multi-scale or cross-level design: the two stages ("global" and "local") operate within the same pyramid level, where the global stage samples a larger neighborhood along the flow and the local stage refines it with a smaller deformable offset, which is not apparent from the figure as the stages look identical. The current terminology gives the false impression of cross-resolution pooling or feature-level aggregation, which is rather a "coarse-to-fine" refinement. Moreover, there is no ablation on each component, confirming their complementary functionality. Further qualitative examples can support the coarse alignment of the first stage and finer attention of the second one.

5. Qualitative example in Figure A.1 fails to convincingly illustrate the benefit of either implicit or explicit mask. The explicit mask has a high weight on both static and dynamic regions visible in both input frames, while the implicit mask appears largely uninformative and conservative, with two sparse islands of higher weights. This raises questions about whether the masking mechanism is functioning as intended or if implementation issues exist. To verify that and further support the claims of complementary mask functionalities, more examples and in-depth analysis have to be provided.

**Questions:**

1. The paper proposes explicit and implicit masks, but it is not clearly explained how the implicit attention map is initialized and learned at the coarsest level, where no upsampled prior exists.
2. No definition of the subscript "_c" in equations (13-14) is provided.
3. The introduction and motivation sections are well-written, but somewhat long, while important discussions (e.g., Table 4 ablations) are relegated to the appendix. This organization forces the reader to flip back and forth between the main text and the appendix. Further reorganization and rewriting of the manuscript would improve readability and focus.
4. There is a typo in line 966: sequence 01 should be stated instead of 00.

---

### Official Review · Reviewer_CRPX · 2025-10-28

**Soundness:** 2
**Presentation:** 2
**Contribution:** 2
**Rating:** 4
**Confidence:** 3

**Summary:**

This paper presents a learning-based visual odometry (VO) framework that integrates optical-flow guidance with deformable correlation and hybrid masking to improve inter-frame correspondences. The method aims to mitigate correspondence failures due to dynamic objects, low-texture regions, and inaccurate correlations in existing VO models. Experiments on KITTI report consistent improvements over prior VO methods, and ablations attribute improvements to both flow-guided deformable correlation and the hybrid masking components.

**Strengths:**

1. Clear problem framing and unified design. The work explicitly targets correspondence instability (dynamic scenes, weak geometric priors) and proposes a unified architecture centered on flow-guided deformable correlation and hybrid masking.
2. Effective alignment via Flow-Guided Deformable (FGD) feature fusion. The FGD feature fusion module integrates optical-flow guidance with learnable deformable offsets in a two-stage hierarchy. This enables flexible, adaptive sampling beyond raw flow, effectively reducing misalignment and yielding higher-quality feature representations.
3. Robust outlier suppression through complementary masks. The hybrid masking strategy combines an explicit flow-consistency mask with a learnable implicit attention mask, effectively suppressing unreliable correspondences while emphasizing informative regions.
4. Consistent gains on KITTI with informative ablations. Results on KITTI show consistent improvements over previous VO methods. Detailed ablations provide supportive evidence that both FGD and hybrid masking contribute to the performance gains.

**Weaknesses:**

1. Insufficient experimental evidence. Because the experiments are confined to KITTI, the study lacks dataset diversity, which weakens the case for the contribution and constrains assessment of robustness across settings. Reporting extra ATE results (e.g., KITTI Seq. 00–06 in Table 3) would strengthen the paper.
2. Missing efficiency metrics. Crucial efficiency metrics (e.g., FPS and memory usage) are absent, complicating assessment of FGD-VO’s practical real-time applicability.
3. Inadequate benchmarking. Because Table 2 benchmarks primarily against dated baselines, it provides an incomplete picture of improvements relative to recent learning-based VO. Including recent learning-based VO baselines would better quantify the performance gains.
4. Figure-level clarity. In Fig. 3, the “patch-wise aggregation” subfigure is confusing and does not clearly convey the multi-point, adaptive sampling process; it does not clearly explain the core FGD mechanism. A redraw with clearer annotations or a more illustrative diagram would be helpful.
5. Terminology consistency. Terminology is inconsistent across sections—Abstract uses “Flow-Guided Deformable Correlation (FGDC) module,” Introduction uses “flow-guided deformable (FGD) correlation module,” Methodology uses “flow-guided deformable (FGD) feature fusion,” Table 4 lists “FGD (removing flow guidance or deformable correlation),” and the Appendix uses “Flow-Guided Deformable (FGD) Module.” Please standardize on a single name and acronym (e.g., FGDC or FGD) and apply it consistently across text, figures, tables, and captions to avoid confusion.
6. Minor presentation issues. In Fig. 2, the branch starting from the image at time T and processed by FlowNet may be misinterpreted as feeding into the image backbone; clarifying arrows/labels would help. In Fig. 3, there is a typo “Gloabl,” which could be corrected to “Global.”

**Questions:**

1. Could you include evaluations on additional datasets (e.g., EuRoC, TUM-RGBD, TartanAir) to strengthen the evaluation?
2. The abstract claims real-time operation, yet efficiency metrics are missing. Could you report end-to-end FPS/latency and GPU memory (with hardware details) to substantiate the real-time claim?
3. Could you clarify why regions such as the sky center receive high hybrid weights in Fig. A.1, or provide additional examples where the hybrid weight more clearly demonstrates its advantage?
4. Given the noticeable ATE improvement on KITTI Seq. 08, could you include the corresponding trajectories in Fig. A.2 to compare predictions with ground truth?
5. Your model resizes inputs to 192×640. Given KITTI’s higher native resolution, could you clarify the rationale for this choice? Was this resolution selected as a trade-off among accuracy and runtime?

---

### Official Review · Reviewer_JNvE · 2025-11-02

**Soundness:** 2
**Presentation:** 1
**Contribution:** 1
**Rating:** 2
**Confidence:** 4

**Summary:**

In this paper, the authors propose to leverage optical flow explicitly for the visual odometry (VO) task.

More specifically, a hybrid masking strategy combing the explicit flow consistency and learnable mask is employed build the confidence of each flow. Besides, a local and global flow refinement process is introduced into a hierarchical pipeline for iterative pose estimation.

Experiments on the single outdoor KITTI datasets demonstrate that the proposed method gives the promising performance  in terms of both rotation and translation estimation.

Ablation studies demonstrate the efficacy of mask, flow guidance, and flow deformation.

**Strengths:**

1.	Good motivation. VO is a fundamental task in computer vision. However, although it has been explored for many years, the current end-to-end frameworks are still suffering from error accumulation partially because of the their sensitivity to dynamic objects and effective usage of geometric information as mentioned in the paper. Therefore, it makes sense that the proposed method uses explicit optical flow as part of input and masks out potential dynamic masks to improve the pose accuracy.


2.	Iterative pose estimation with global and local optical flow refinement. The higher resolutions of optical flow map contain richer geometric details, which may predict more precise camera poses, so the iterative pose estimation by leveraging features from low to high resolutions sounds good. Besides, the global and local optical flow refinement should be able to give more accurate optical flow, resulting in more accurate poses.

3.	The idea is simple and easy to follow, though applying optical flow to VO framework is not new.

**Weaknesses:**

There are several weaknesses of this paper.

1.	Presentation of the paper. The methodology section is not clear at all.

According to line 189-193, the flow net should accept consecutive frames as input. However, Figure 2 shows that Flow net only accepts a single frame as input.

Section 3 gives many notations used in the pose estimation process, but none of them can be found in Figure 3, reducing the readability of the paper. I would suggest the authors putting important notations also on Figure 2 to help readers better understand the whole method.


2.	Unclear description of the method. Some notations in the section do not have clear description. For example, how to get the attention map A^{l+1} in Eq (7), how to get the positional embedding E^{l} in Eq (13)? Some of these are included in the appendix, but the missing description causes a huge difficulty of understanding the method. I would recommend reorganizing the methodology section thoroughly.

3.	Evaluation. The proposes method is only evaluated on the single outdoor dataset – kitti, which is not sufficient to prove the effectiveness of the proposed approach. Moreover, the proposed method even gives worse performance than some prior approaches e.g. [R1].

I think the evaluation on the indoor datasets such as TUM-RGBD would also be necessary.


[R1] Deep visual odometry with adaptive memory, Xue et al., TPAMI 2020.

**Questions:**

Please see the weaknesses section.

---

### Note · Authors · 2025-11-12

I have read and agree with the venue's withdrawal policy on behalf of myself and my co-authors.